# Avoiding Overfitting to the Importance Weights in Offline Policy Optimization

## Abstract

Offline policy optimization has a critical impact on many real-world decision-making problems, as online learning is costly and concerning in many applications. Importance sampling and its variants are a widely used type of estimator in offline policy evaluation, which can be helpful to remove assumptions on the chosen function approximations used to represent value functions and process models. In this paper, we identify an important overfitting phenomenon in optimizing the importance weighted return, and propose an algorithm to avoid this overfitting. We provide a theoretical justification of the proposed algorithm through a better per-state-neighborhood normalization condition and show the limitation of previous attempts to this approach through an illustrative example. We further test our proposed method in a healthcare-inspired simulator and a logged dataset collected from real hospitals. These experiments show the proposed method with less overfitting and better test performance compared with state-of-the-art batch reinforcement learning algorithms.

## 1 Introduction

Learning decision policies from offline data has a critical impact on real-world decision-making applications such as healthcare (Lei et al., 2012; Komorowski et al., 2018), recommendation systems (Li et al., 2010; Chen et al., 2019), and intelligent tutoring systems (Mandel et al., 2014).In real-world decision-making applications, we often need to use limited amounts of data about past decisions to yield better future performance in complex, hard-to-model domains. Such offline policy optimization encounters two key generalization challenges: distribution shift and modeling bias. Distribution shift arises because, under a new decision policy, the distribution of observations and actions will be different than the distribution of observations and actions in the offline data. Modeling bias can occur when popular assumptions over the domain structure or model class (such as the Markov assumption, realizability, or the value function class being closed under the Bellman), fail to hold in the problem setting. Learning from limited data, algorithms ignoring these challenges can produce policies and estimates of policy performance that are far more optimistic than they achieve in practice, similar to the overfitting in supervised learning.

To address this, prior work in batch reinforcement learning (RL) and contextual bandits mostly falls into two categories: (1) changing the estimator or loss function used in computing offline value estimates, or (2) constraining the policy search space. In category (1), self-normalized IS (or weighted IS) (Hesterberg, 1995; Swaminathan & Joachims, 2015b), variance regularization (Swaminathan & Joachims, 2015a; Metelli et al., 2018), doubly robust methods(Dudík et al., 2011; Jiang & Li, 2016; Thomas & Brunskill, 2016), and uncertainty regularized model-based batch RL (Yu et al., 2020; Kidambi et al., 2020) are all approaches that aim to provide more stable and variance-penalized estimates of a policy's performance to address the uncertainty due to limited data and distribution shift. The second category of direct constraints on the policy class includes bandit learning under action/policy constraints with behavior probability (Sachdeva et al., 2020), value-based batch RL algorithms by constraining the policy disagreement (Kumar et al., 2019; Buckman et al., 2020), thresholding the action probability (Fujimoto et al., 2019; Futoma et al., 2020), or thresholding the marginalized state-action probability (Liu et al., 2020), under the behavior policy.

Many of the works above rely on learning MDP models or value functions based on MDP assumptions. However, the Markov assumption and function approximation assumptions are often unrealistic in

real-world problems and hard to verify, which limits their eligibility in certain application problems. In this paper, we consider learning an policy to optimize an importance weighted return, as it is free from Markov and other types of model assumptions. We identify an unique overfitting that is not solved in prior work about policy optimization and propose a new constraint on policy named *eligible actions* (EA) to address the overfitting issue.

The overfitting issue comes from fitting the importance weights in the weighted return objective. The importance weighted return can effectively remove a certain subgroup of samples in the dataset by decreasing all weights to zero. This context avoidance is unrealistic when the subgroup consists of a subset of initial states, as the policy does not get to control the initial state distribution. Consider the following motivating example: an AI decision policy may learn from historical treatment logs to avoid all the logged treatment in the history for patients with severe conditions as they lead to low rewards. This avoidance can be achieved with finite data if the action space is large, and such avoidance will result in high off-policy estimates. However, in the real environment, a policy still should make good decisions for patients who arrive with severe conditions. This *propensity overfitting* problem was studied in contextual bandit problems (Swaminathan & Joachims, 2015b). While their solution effectively constrained the upper bound on importance weights on high-reward samples, the challenge of overfitting to avoid the low-reward samples remains.

**Our contribution** To address the issue that optimizing with propensity weights may avoid certain initial states, we propose the **P**olicy **O**ptimization with **EL**igible **A**ctions (POELA) algorithm. We constrain the potential action set to the set of observed actions with similar states to prevent improper avoidance to lower-reward initial states. Thus within a radius of a particular state, the sum of observed importance weights are always lower bounded. We then study the empirical performance of POELA in a medical simulator and a real-world medical dataset. Our approach finds much better decision policies than prior state-of-the-art algorithms in both domains.

## 2 OFFLINE POLICY OPTIMIZATION

We study the problem of offline policy optimization in sequential decision-making under uncertainty. Let the environment be a finite-horizon Contextual Decision Process (CDP) (Jiang et al., 2017). A CDP can capture more general, non-Markovian settings (also sometimes refered to as a Non-Markov DP (Kallus & Uehara, 2019b)). A CDP is defined as a tuple $\langle \mathcal{X}, \mathcal{A}, H, P, R \rangle$, where $\mathcal{X}$ is the context space, $\mathcal{A}$ is the action space, and $H$ is the horizon. $P = \{P_h\}_{h=1}^H$ is the unknown transition model, where $P_h : (\mathcal{X} \times \mathcal{A})^{h-1} \to \Delta(\mathcal{X})$ is the distribution over next context given the history $P_1 : \Delta(\mathcal{X})$ is the initial context distribution. Similarly, $R = \{R_h\}_{h=1}^H$ is the reward model and $R_h : (\mathcal{X} \times \mathcal{A})^h \to \Delta([-R_{\max}, R_{\max}])$.

In this paper we focus on learning policies that map from the most recent context to an action distribution, $\pi : \mathcal{X} \to \Delta(\mathcal{A})$. This is optimal when the domain is Markov and can often be more interpretable and more feasible to optimize given finite data in the offline setting. In offline policy optimization settings, we have a dataset with n trajectories collected by a fixed *behavior* policy $\mu : \mathcal{X} \to \Delta(\mathcal{A})$, and we aim to find a policy $\pi$ in a policy class $\Pi$ with the highest value.

Policy gradient and optimization approaches do not rely on a Markov assumption on the underlying domain, and have had some encouraging success in offline RL (Chen et al., 2019). Often these methods leverage an importance sampling (IS) estimator in policy evaluation: $\hat{v}_{\mathrm{IS}}(\pi) = \frac{1}{n} \sum_{i=1}^n \left( \sum_{h=1}^H r_h^{(i)} \right) \prod_{t=1}^H \left( \frac{\pi(a_h^{(i)}|x_h^{(i)})}{\mu(a_h^{(i)}|x_h^{(i)})} \right)$. The IS estimator is an unbiased and consistent estimate of the value under the following two assumptions:

**Assumption 1** (Overlap). *For any $\pi \in \Pi$, and any $x \in \mathcal{X}$, $a \in \mathcal{A}$, $\frac{\pi(a|s)}{\mu(a|s)} < \infty$.*

**Assumption 2** (No Confounding / Sequential ignorability). *For any policy $\pi \in \Pi$ and $\mu$, conditioning on the current context $x_h$, the sampled action $a_h$ is independent of the outcome $r_{h:H}$ and $x_{h+1:H}$.*

IS often suffers from high variance, which has prompted work into extensions such as doubly robust methods (Jiang & Li, 2016; Thomas & Brunskill, 2016) and/or methods that balance variance and bias. Truncating the weights and using self-normalization has been shown to be empirically beneficial both in bandit and RL settings (Swaminathan & Joachims, 2015b; Futoma et al., 2020): we refer to

this as Self-Normalized Truncated IS (SNTIS):

$$\hat{v}_{\text{SNTIS}}(\pi) := \frac{1}{\sum_{i=1}^n \min\left\{\prod_{t=1}^H W_h^{(i)}, M\right\}} \sum_{i=1}^n \left(\sum_{h=1}^H r_h^{(i)}\right) \min\left\{\prod_{t=1}^H W_h^{(i)}, M\right\}, \quad (1)$$

where $W_h^{(i)} := \frac{\pi(a_h^{(i)}|x_h^{(i)})}{\mu(a_h^{(i)}|x_h^{(i)})}$. For the ease of notation, in the rest of this paper we define $W_{1:h}^{(i)} := \prod_{t=1}^h W_t^{(i)}$, $W^{(i)} := W_{1:H}^{(i)}$, $W = \sum_{i=1}^n W^{(i)}$, and $r^{(i)} = \sum_{h=1}^H r_h^{(i)}$.

## 3 THE OVERFITTING PROBLEM IN IMPORTANCE WEIGHTED POLICY OPTIMIZATION

We now highlight an important problem with using important sampling estimators such as the above during offline policy optimization. For concreteness we will describe this in the contextual bandit setting using the self-normalized importance sampling estimator, but a similar results hold for un-normalized estimator. We will also shortly describe in the main text how this issue persists in existing importance weighted off-policy learning methods and about the multi-step case in the appendix.

The key issue is that given a finite dataset, during optimization the IS estimator objective can benefit policies that improperly avoid low reward states, inducing a lack of overlap on the *empirical* behavior data. To see this, let $v^\pi(x) = \mathbb{E}[r|x, a \sim \pi]$, $\hat{p}(x)$ be the empirical probability mass/density over the contexts $x$ in the dataset, and $W(x) = \sum_{i:x^{(i)}=x} \frac{W^{(i)}}{W}$. We now decompose the importance weighted off-policy estimator into three parts.

$$\hat{v} = \underbrace{\mathbb{E}_{\hat{p}}[v^\pi(x)]}_{\text{empirical } v} + \underbrace{\sum_{x \in \mathcal{X}} (\hat{p}(x) - W(x))v^\pi(x)}_{\text{difference in context weights}} + \underbrace{\sum_{x \in \mathcal{X}} W(x) \left(\sum_{x^{(i)}=x} \frac{W^{(i)}}{W(x)W}r^{(i)} - v^\pi(x)\right)}_{\text{weighted IS error in each context}} \quad (2)$$

The first term is a supervised empirical value estimate whose only error is due to the error in the empirical context distribution sampled in the dataset, vs the true context distribution. The second term captures the error caused by the difference between context distribution introduced by weights and empirical context distribution in the dataset. The third term computes the difference between the weighted IS estimate of the value of the policy in a specific context $x$ versus its true value $v^\pi(x)$, and then sums this over all contexts.

The second term is of particular interest, because it highlights how the IS estimator of a policy may effectively shift the relative weight on the context space. Note in the bandit setting (and in the initial starting state distribution for the RL setting) such shifting should not be allowed: the policy may control what actions are taken, but cannot change the initial context distribution.

Before illustrating the problematic behavior that can result from this type of objective function for offline policy learning, we first further analyze if it is possible for $W(x)$ to differ significantly from $hatp(x)$, given the normalization constraint of $\pi$. Expanding the difference between the empirical distribution over a context and $W(x)$:

$$\hat{p}(x) - W(x) = \hat{p}(x) - \sum_{i:x^{(i)}=x} \frac{W^{(i)}}{W} = \hat{p}(x) - \frac{N * \hat{p}(x)}{W} \sum_a \hat{p}(a|x) \frac{\pi(a|x)}{\mu(a|x)} \quad (3)$$

$$= \left(1 - \frac{N}{W}\right)\hat{p}(x) + \frac{N * \hat{p}(x)}{W} \sum_a \left(1 - \frac{\hat{p}(a|x)}{\mu(a|x)}\right)\pi(a|x) \quad (4)$$

While the first term is asymptotically zero because $W \to N$, the second term will be non-zero whenever the policy $\pi$ puts non-zero weight on actions in contexts where the empirical behavior distribution differs significantly from the true behavior distribution. Such a difference between the empirical and true behavior distributions can easily happen. This is especially the case in settings where the state-action space is large and the number of observed actions per state is small, which is common in recommendation settings, healthcare and many other applications.

A policy learning algorithm maximizing the objective in Equation 2 can exploit this difference in context weights and return a policy with an over inflated value estimate, as we now illustrate:

**Example 1.** *Consider a contextual bandit problem with $|\mathcal{X}|$ contexts and $|\mathcal{A}|$ actions in each context. For half the contexts $S_p$, the reward is 1 for one action and zero for others. For the other half of the contexts, $S_n$, the reward is -1 for half the actions, and -5 for the rest. The true distribution over contexts is uniform. Note in this setting the optimal policy would have a reward of 0. The behavior dataset is draw from a uniform distribution over contexts and actions. When the sample size $|\mathcal{D}| < |\mathcal{A}|$, we can assume there is only one observed positive reward in the dataset. Then computing the policy to maximize the important sampling weighted objective will yield weights that are 0 on all contexts except for the context with the positive reward, through a policy $\pi$ that selects actions that are not present in the dataset for all contexts whose observed actions lead to only zero or negative rewards: let $\mathcal{A}_{s_i} = \{a_i : r(s_i, a_i) \leq 0\}$ then $\pi(s_i) = a_j$ where $a_j \notin \mathcal{A}_{s_i}$.*

*The resulting IS/SNIS estimator of the value of $\pi$ is 1, which is much higher than the optimal policy value. In addition, note that the returned policy $\pi$ is likely to be worse than the optimal policy for any context where $r(s_i, a_i) = -1$, since that is the optimal reward possible for such states $s_i$, and by $\pi$ selecting an unobserved action $a_j$ in that state ($\pi(s_i) = a_j$), the policy $\pi$ may select an action with worse true reward, $r(s_i, a_j) = -5$.*

Note that in this example, there is no error in the importance weighted return for the contexts with positive $W(x)$ (which is the third term in the decomposition shown in Equation 2). The issue arises because to optimize the objective, it is possible to select a policy whose $W(x)$ distribution over the contexts has zero weight on at least the half of the contexts with negative rewards $S_n$, by assigning them actions not selected in the dataset. This causes the second term in the decomposition (Equation 2) to be significantly overestimated.

This is a unique overfitting phenomena in the counterfactual (or called off-policy) learning settings since in supervised learning it is not possible for a hypothesis $\pi$ to manipulate the weights over the input contexts. While this issue can arise in contextual bandits with large state and action space in small datasets, we show that it is even easier for this to occur in sequential reinforcement learning settings, even when only 2 actions are available in Example 2 and 3 in the Appendix.

Intuitively, the issue arises because in estimating the value of a new decision policy, it is acceptable to choose a policy that re-distributes the *weights of actions within an initial context* but not re-distribute the *weights across initial contexts*, since it is not a function of the actions selected. It is well known that in importance sampling, the expected ratio of the weights should be 1: $\mathbb{E}_{y \sim \mu}[\pi(y)/\mu(y)] = 1$. In contextual policies, we expect that the expected weights should also be 1 for all initial contexts $x_0$: $\mathbb{E}_{a \sim \mu(a|x_0)}[\pi(a|x_0)/\mu(a|x_0)|x_0] = 1$. However, optimizing for a standard importance sampling objective (such as Equation 2) does not involve constraints that the empirical expectation of weights given an initial context $\hat{\mathbb{E}}\left[W_h^{(i)}|x_h^{(i)})|x_h^{(i)}\right]$ (or the weights of $n$-step given initial context $\hat{\mathbb{E}}\left[W^{(i)}|x_1^{(i)})|x_1^{(i)}\right]$ ) is still close to one. In order to optimize the future return, the optimization algorithm can optimize the weights of initial contexts in the dataset, instead of optimizing the weights on different actions/ action sequences in a given initial context.

This result may seem surprising, given that under mild assumptions which are satisfied here (Assumptions 1 and 2) importance sampling is well known to provide an unbiased estimate of the value of a policy. Our observations do not contradict this fact: while importance sampling will still provide an unbiased estimate given a policy, the policy optimization can exploit the finite sample error and its heterogeneousness across the policy class.

## 4   PRIOR WORK ON OFFLINE POLICY OPTIMIZATION

Before we describe a method to alleviate this problem, we first review prior work. There is increasing interest in multi-armed bandits and offline RL to avoiding overly optimistic estimates of policies computed from finite datasets that can cause suboptimal policy learning, and prior work has discussed extrapolation error(Fujimoto et al., 2019), propensity overfitting (Swaminathan & Joachims, 2015b), and bandit error (Brandfonbrener et al., 2020). The problem we identified arises specifically from deficient support in the observed dataset.

To address potential overfitting, one line of work focuses on changing the objective used for policy optimization. For example, Swaminathan & Joachims (2015b) has shown that policy optimization

using vanilla importance sampling in contextual bandits will over-maximize the weights if the reward is positive and over-minimize the weights for negative reward. Shifting the rewards will help with the latter (and avoid part of the context avoidance we describe) but can worsen the former. The self-normalized estimator proposed to address the issue of over-maximizing reward (Swaminathan & Joachims, 2015b) is equivariant to any constant shift in rewards, and will still suffer from the context avoidance issues we describe above. Counterfactual risk minimization (Swaminathan & Joachims, 2015a) uses variance regularization based on the empirical Bernstein's inequality. However, this penalization is at the policy level and does not directly address the problem with avoiding contexts with low reward. In Appendix Figure 2c we show the counterfactual risk minimization regularization with or without self-normalization requires a large dataset to perform well. Later work (Joachims et al., 2018) extended norm-POEM to stochastic gradient descent settings for large-scale training. Recent work (Brandfonbrener et al., 2020) discussed a similar overfitting issue as we describe and compared the performance of offline policy optimization and model/value-based method on this overfitting issue. Those authors focus primarily on the negative result of the policy optimization approach and the advantage of the model/value-based method. Doubly robust estimators (Dudík et al., 2011; Jiang & Li, 2016; Thomas & Brunskill, 2016; Kallus & Uehara, 2019a;b) have multiple benefits but as long as the learned $Q$ function is imperfect, policy learning can still overfit to the high/positive residual $r - Q$. Pessimism under uncertainty approaches are promising (Kidambi et al., 2020; Yu et al., 2020) but have so far only been developed for Markov settings and are not robust to model class misspecification.

Another line of offline batch policy optimization constrains the policy search space, for example to constrain target policies to be close in some distance to the behavior policy (Kumar et al., 2019; Buckman et al., 2020). A common approach is to constrain target policies to have a minimum conditional action probability under the behavior policy $\mu(a|s)$ (e.g. Sachdeva et al. (2020); Fujimoto et al. (2019); Futoma et al. (2020); Liu et al. (2020)). This work has focused on algorithms and analysis for the Markov setting with additional model realizability assumptions, except Sachdeva et al. (2020) in the bandit case and Futoma et al. (2020) in the RL setting. For example, the MDP value based approaches often require the value function class is closed respect to the Bellman operator and the value function is realizable. Such assumptions are hard to verify in real-world domains.

This prior work does not address the issues we outlined, that occur due to deficient support in the observed finite data rather than in the expected behaviors. Our work can be broadly viewed as following in the recent line of work on pessimism under uncertainty, but adapted to provide policy search based offline learning method that does not require the Markov assumption or model realizability assumptions, and can achieve strong performance given a finite dataset.

## 5 Lower Bound of $\hat{\mathbb{E}}[\mathbf{W}|\mathbf{x}]$ and Eligible Actions

We have observed that the issue rises in policy learning because weight can be placed on unobserved actions for certain contexts, causing the empirical conditional expectation of weights given a context to be zero for such contexts. To address this, one possibility is to constrain the conditional expectation of weights given a context to be 1 or lower bounded. However doing so in infinite/continuous context spaces is subtle: each context likely only appears once in the dataset, and requiring $\hat{\mathbb{E}}[W|x] = 1$ would be equivalent to only allowing a policy that exactly matches the observed logged actions.

We now propose a slight relaxation of the above proposal. Intuitively the issue arises because without further constraints, policy learning can place large weight on unobserved actions in the dataset, for which our reward uncertainty is high. The recent line of pessimism under uncertainty for model and value based MDP offline RL explicitly accounts for such statistical uncertainty through constraining or penalizing actions and/or states and actions for which there are limited observed data. Similar to this, we introduce a pessimistic constraint on the policy class to be considered, and then demonstrate this allows us to constrain the empirical conditional expectation of weights given a context.

For policy learning, we create local constraints on the eligible policy class by defining for each context $x$, dataset $\mathcal{D}$ and a given threshold $\delta$, the *eligible action* set $A(x; \mathcal{D}, \delta)$

$$A_h(x; \mathcal{D}, \delta) = \{a_h : \exists (x_h, a_h) \in \mathcal{D} \ s.t. \ \text{dist}(x, x_h) \leq \delta\} \tag{5}$$

Intuitively this allows any action that was taken for a given context, or actions taken in contexts within a given distance of the observed contexts.

We now show that any policy that only selects actions in eligible action sets will ensure that the empirical sum of weights in any hypersphere of any context can be lower bounded, as desired. To do so we first introduce an assumption about the policy's smoothness in the context space.

**Assumption 3** ($L$-Lipschitz policy). $\forall \pi \in \Pi$, $\|\pi(a|x) - \pi(a|x')\| \leq L\mathrm{dist}(x, x')$.

Under this and the former assumptions, we can now show the following. All proofs, when omitted, are provided in the appendix:

**Theorem 1.** $\forall x_h^{(i)}$, $\mathcal{B}(x_h^{(i)}, \delta) := \{x : \mathrm{dist}(x, x_h^{(i)}) \leq \delta\}$, $\sum_{x_h^{(j)} \in \mathcal{B}(x_h^{(i)}, \delta)} W_h^{(j)} \geq 1 - \delta L|\mathcal{A}|$

Given the likelihood ratio is lower bounded, we can further show that the self-normalized truncated weights are also lower bounded in the one-step settings.

**Corollary 1.** For $H = 1$, $\sum_{x_1^{(j)} \in \mathcal{B}(x_1^{(i)}, \delta)} \frac{\max\{W^{(i)}, M\}}{\sum_{i=1}^n \max\{W^{(i)}, M\}} \geq \frac{1 - \delta L|\mathcal{A}|}{nM}$ for $M > 1$.

In $n$-step sequential settings, it is necessary to have the 1-step weights be greater than zero in order to have $n$-step weights greater than zero.

**Proposition 1.** For any $x$, $\delta$, $\mathbb{E}[W_{1:h}^{(i)}|x_h^{(i)} \in \mathcal{B}(x, \delta)] = \mathbb{E}[W_{1:h-1}^{(i)}|x_h^{(i)} \in \mathcal{B}(x, \delta)]\mathbb{E}[W_h^{(i)}|x_h^{(i)} \in \mathcal{B}(x, \delta)]$. $\hat{\mathbb{E}}[W_{1:h}^{(i)}|x_h^{(i)} \in \mathcal{B}(x, \delta)] = \hat{\mathbb{E}}[W_{1:h-1}^{(i)}|x_h^{(i)} \in \mathcal{B}(x, \delta)]\hat{\mathbb{E}}[W_h^{(i)}|x_h^{(i)} \in \mathcal{B}(x, \delta)]$

If the policy overfits the weights on an h-step context such that $\hat{\mathbb{E}}[W_h^{(i)}|x_h^{(i)} \in \mathcal{B}(x, \delta)] = 0$, the $n$-step weights will also be zero even the roll-in probability under $\pi$ is non-zero.

The above results highlight that introducing a constraint on the policy class to have local overlap with actions taken in the dataset, is sufficient to ensure that the weights on the contexts are lower bounded. This will help address the overfitting issue highlighted in the prior sections. Policy learning can now be done by finding the policy which satisfy the eligible actions constraints given the input dataset:

$$\arg\max_{\pi \in \Pi} J(\pi; \mathcal{D}) \quad s.t. \forall i, h \sum_{a \in A_h(x_h^{(i)}; \mathcal{D}, \delta)} \pi(a|x_h^{(i)}) = 1 \tag{6}$$

$J(\pi; \mathcal{D})$ can be any objective function such as $\hat{v}_{\mathrm{IS}}$, $\hat{v}_{\mathrm{SNTIS}}$ or with the counterfactual risk minimization regularization (Swaminathan & Joachims, 2015a) in prior work:

$$\arg\max_{\pi \in \Pi} \hat{v}_{\mathrm{SNTIS}}(\pi) - \lambda\sqrt{\widehat{\mathrm{Var}}(\hat{v}_{\mathrm{SNTIS}}(\pi))}. \tag{7}$$

A natural question is whether these action eligibility local constraints limits the expressivity of the policy class $\Pi$. We show that asymptotically the expressivity is the same, and the maximizer in Equation 6 will converge to the optimal policy in the policy class if $J(\pi, \mathcal{D})$ is consistent.

**Theorem 2.** *i) Fixed $\delta$, for any $x$, $A_h(x; \mathcal{D}, \delta) \to \{a : \mu(a|x) > 0\}$ as $n \to \infty$ with probability 1. Thus the solution to Equation 6 is the same as $\hat{\pi}_{\mathcal{D}, J} = \arg\max_{\pi} J(\pi, \mathcal{D})$. ii) If $J(\pi, \mathcal{D})$ is the objective in Equation 7, the truncation threshold $M$ as a function satisfies $M \to \infty$ and $M/n \to 0$ as $n \to \infty$, and $|\Pi| < \infty$, then $v^{\hat{\pi}_{\mathcal{D}, J}} \to \max_{\pi \in \Pi} v^{\pi}$ in probability.*

# 6 ALGORITHM: POLICY OPTIMIZATION WITH ELIGIBLE ACTIONS

---

**Algorithm 1** Policy Optimization with ELigible Actions (POELA)

---

1: **Input:** $\mathcal{D}$, $\Pi_\theta$, sphere radius $\delta$, IS truncation $M$, CRM coefficient $\lambda$, learning rate $\alpha$
2: **Output:** $\hat{\pi}_\theta$
3: Initialize $\theta_0$
4: **for** $t = 0, 1$ **until convergence do**
5: $\quad \hat{\pi}_{\theta_t}(a|x) := \mathbb{1}\{a \in A_h(x; \mathcal{D}, \delta)\}\pi_{\theta_t}(a|x) / \left(\sum_a \mathbb{1}\{a \in A_h(x; \mathcal{D}, \delta)\}\pi_{\theta_t}(a|x)\right)$
6: $\quad \theta_{t+1} \leftarrow \theta_t + \alpha\nabla_\theta\left(\hat{v}_{\mathrm{SNTIS}}(\hat{\pi}_{\theta_t}) - \lambda\sqrt{\widehat{\mathrm{Var}}(\hat{v}_{\mathrm{SNTIS}}(\hat{\pi}_{\theta_t}))}\right)$
7: **end for**

---

Now in Algorithm 1 we introduce our POELA (Policy Optimization with Eligible Actions) algorithm that implements the learning objective in Equation 6 (see Algorithm1—. We use the counterfactual

risk minimization objective function Equation 7 as the $J(\pi; \mathcal{D})$, where the estimator $\widehat{\text{Var}}(\hat{v}_{\text{SNTIS}})$ is constructed using the Normal approximation in [ (Owen, 2013, Equation 9.9) $\widehat{\text{Var}}(\hat{v}_{\text{SNTIS}}) = \frac{\sum_{i=1}^{n}\left(r^{(i)} - \hat{v}_{\text{SNTIS}}\right)^2 (\min\{W^{(i)}, M\})^2}{\left(\sum_{i=1}^{n} \min\{W^{(i)}, M\}\right)^2}$. After each gradient step, we enforce the policy to satisfy the eligible action constraints by re-normalizing the output probability on $A_h(x; \mathcal{D}, \delta)$ for $x \in \mathcal{D}$. The eligible action set for each training sample is static and can be stored to reduce computational cost. In the experiments, we use Euclidean distance in over nearby states at any time index.

# 7 EXPERIMENT

We now compare POELA with several prior methods for offline RL. Perhaps the most relevant work in avoiding overfitting when using importance sampling is norm-POEM (Swaminathan & Joachims, 2015b). Here we extend it here to be suitable for sequential decision settings, use a neural network policy class and refer to the resulting algorithm as PO-CRM. A second baseline PO-$\mu$ constrains the policy class to only include policies which take actions with a sufficient probability under the behavior policy $\mu(a|s)$ (see e.g.Futoma et al. (2020)). We also compared with recent deep value-based MDP methods in batch RL: BCQ (Fujimoto et al., 2019) and PQL (Liu et al., 2020). For all algorithms we use a feedforward neural network for the relevant policy and/or value, function approximators.

We use the same procedures for each algorithm to select its hyper-parameters. An algorithm is trained on the training set multiple times, using different hyper-parameters and several restarts. Intermittent policies generated during the training process are saved at checkpoints. Across this set of potential policies, we identify the policy with the highest self-normalized truncated IS (SNTIS) estimates on a held-out validation set. Finally, we report the test performance of selected policy either through online Monte-Carlo estimation if a simulator is available, or using SNTIS estimates on a held out test set. Full experimental details are provided are in the Appendix.

## 7.1 EXPERIMENT IN LGG TUMOR GROWTH INHIBITION SIMULATOR

The Tumor Growth Inhibition (TGI) simulator (Ribba et al., 2012) describes low-grade gliomas (LGG) growth kinetics in response to chemotherapy in a horizon of 30 steps (months), with an non-Markov context and a binary action of drug dosage (Yauney & Shah, 2018). The reward consist of an immediate penalty proportional to the drug concentration, and a delayed reward of the decrease in mean tumor diameter. The behavior policy selects from a fixed dosing schedule of 9 months (the median duration from Peyre et al. (2010)) with 70% probability and else selects actions at random.

In this experiment, the behavior policy can only take values in $\{0.15, 0.85\}$. This means constraining the policy class to have a minimal probability under $\mu(a|s)$, as in baseline PO-$\mu$, is only a non-trivial constraint for thresholds greater than 0.15: this produces a single potential target policy, which is the deterministic fixed-dosage part of behavior policy. We include this as 9-mon (short for 9 month dosing) in Table 1. The training and validation set both have 1000 episodes, and we repeat the experiment 5 times with 5 different train and validation sets. Policy values are normalized between 0 (uniform random) and 100 (best policy from online RL).

As shown in Table 1 (Non-MDP rows), our POELA achieves the highest test value as well as smaller variability compared with the baselines.

**Does POELA reduce overfitting?** We calculate the difference between $\hat{v}_{\text{SNTIS}}$ on the validation set and the online test value. Although for a given algorithm, we select the final policy using off-policy evaluation on the validation set, most algorithms still result in a policy whose value is a significant overestimate of its true performance (see Table1, rows for $\hat{v}_{SNTIS} - v^\pi$). In contrast, our approach yields a policy whose value is much more accurately estimated and performs better.

**Performance comparison in MDP environment** We also repeat the experiment with an MDP modification of the simulator, including an immediate Markovian reward and additional features for a Markovian state space. Note that we expect BCQ and PQL to do very well: both are designed to avoid overfitting in offline MDP learning and in particular PQL uses a pessimism under uncertainty approach to penalize policies that put weight on state–action pairs with little support. Though our POELA algorithm makes no Markov assumptions, it performs only slightly worse than the conservative MDP methods. POELA also substantially outperforms other policy-based approaches.

|  | Algorithms | POELA | PO-CRM | BCQ | PQL | 9-mon |
|---|---|---|---|---|---|---|
| Non-MDP | Test $v^\pi$ | $92.20 \pm 1.63$ | $75.06 \pm 13.22$ | $30.24 \pm 15.22$ | $74.76 \pm 9.75$ | 68.12 |
|  | $\hat{v}_{\text{SNTIS}} - v^\pi$ | $-1.26 \pm 1.92$ | $15.57 \pm 13.07$ | $63.72 \pm 15.54$ | $17.74 \pm 9.49$ | $-$ |
| MDP | Test $v^\pi$ | $89.52 \pm 1.55$ | $78.79 \pm 6.42$ | $95.64 \pm 4.64$ | $96.88 \pm 3.76$ | 68.12 |
|  | $\hat{v}_{\text{SNTIS}} - v^\pi$ | $5.16 \pm 1.78$ | $14.93 \pm 5.71$ | $-0.69 \pm 0.81$ | $0.55 \pm 5.12$ | $-$ |

Table 1: Test $v^\pi$ and amount of overfitting of the learned policy. Test $v^\pi$ is computed from 1000 rollouts in the simulator. $\hat{v}_{\text{SNTIS}}$ on the validation set $-$ test $v^\pi$ represents the amount of overfitting. All numbers are averaged across 5 runs with the standard error reported.

| Method | POELA | PO-$\mu$ | PO-CRM | BCQ | PQL | Clinician |
|---|---|---|---|---|---|---|
| Test SNTIS | 91.46 (90.82) | 87.95 | 87.71 | 82.67 | 84.40 | 81.10 |
| 95% BCa UB | 93.24 (92.61) | 90.58 | 90.04 | 86.83 | 88.29 | 82.19 |
| 95% BCa LB | 89.59 (88.68) | 84.77 | 84.90 | 78.25 | 80.13 | 79.80 |
| Test ESS | 624.92 (586.37) | 372.00 | 399.59 | 228.82 | 231.93 | 2995 |

Table 2: MIMIC III sepsis dataset. Test evaluation, $(0.05, 0.95)$ BCa bootstrap interval, and effective sample size. The value of POELA without a CRM variance penalty is shown in parentheses.

## 7.2 EXPERIMENT IN ICU DATA - MIMIC III

We next apply our method in a real-world example of learning policies for sepsis treatment in medical intensive care units (ICU). We use an extracted cohort (Komorowski et al., 2018) of patients fulfilling the sepsis-3 criteria from the MIMIC III data set (Johnson et al., 2016) and obtain a dataset of 14971 patients, 44 context features, 25 actions and a 20 step maximum horizon. Full details are in the Appendix. We hold out 20% of data for validation and 20% of data for the final test. The treatment logs do not include the probabilities of clinicians' actions. Instead, as suggested by prior work(Raghu et al., 2018), we estimate the probabilities of the behavior clinicians' policy by $k$-NN with $k = 100$. To ensure overlap, for all policy optimization algorithms we allow $\pi(a|s) > 0$ only if $\hat{\mu}(a|s) > 0$.

Using self-normalized truncated importance sampling to evaluate the performance on a test set is appealing because it makes little assumptions on the underlying domain. However, it may be that very few test behavior policy trajectories match a candidate test policy, which can make the resulting value estimate unreliable. The amount of overlap between the test set and a desired policy to evaluate can be measured quantitatively by the effective sample size (ESS) (Owen, 2013). To help increase the chance that the test effective sample size is sufficient to yield reliable estimates, during the policy selection process for a given algorithm, only policies with an effective sample size of at least 200 on the validation set are considered.[1] Similar to prior work (Thomas et al., 2015), in addition to the SNTIS estimator on the test set, we also report a 95% upper and lower bound from bias-corrected and accelerated (BCa) bootstrap. The clinician's column is the test dataset rewards and sample size.

Table 2 shows our POELA is the best on all metrics, achieving the highest evaluation on the test set, the highest upper and lower bounds, and the highest effective sample size.

**Is the variance regularization helpful?** To demonstrate the effect of variance penalty on the POELA algorithm, we demonstrate the test performance POELA without the variance penalty. The performance is worse than using the variance penalty but still higher than the baseline algorithms.

**Is there a trade-off between effective sample size (ESS) and performance estimates?** A tension in conservative offline optimization is that the most reliable and conservative policy estimates come from effectively imitating the behavior policy (which will maximize effective sample size). Policies that differ substantially from the behavior policy may yield higher performance, but have less overlap with the existing logged data (and lower ESS). In Figure 1a we plot each hyper-parameter and each re-start from different algorithms. The plots show that POELA achieve a better Pareto frontier between performance estimates and effective sample size than other algorithms.

---

[1]Note that the variance penalty may not ensure that the ESS is large. In particular, if only 2 trajectories in a dataset match a desired target policy (an ESS of 2). When they have the same reward and weights, the variance penalty of self-normalized estimator will be 0.

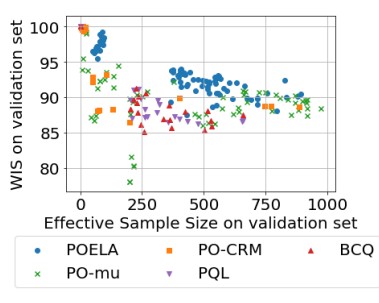

(a) Trade-off between ESS and OPE

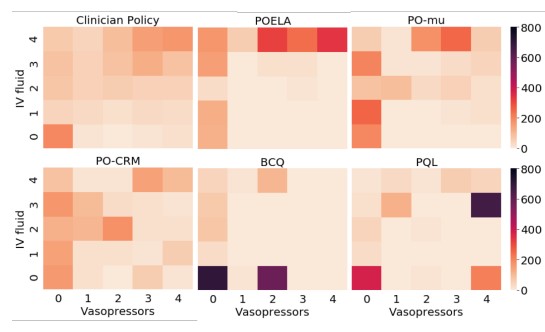

(b) Action counts in high-SOFA contexts

Figure 1: (a): Trade-off between effective sample size and value estimates. (b): Action counts heatmap in high-SOFA initial contexts of the policy learned.

**Do learned policies take medically reasonable actions on sicker patients?** As we discuss, prior IS methods can overfit to achieve high estimated IS/SNIS score by avoiding initial contexts with low expected rewards. We now explore the learned policies for patients with high logged SOFA score (measuring organ failure) initially in the test dataset. Figure 1b depicts the number of actions that would be taken by different policies, as well as by the clinicians on the high-SOFA contexts. POELA mainly takes high-vasopressors high-IV-fluid treatments but also some IV-fluid-only treatments, which is similar to the clinician's policy but more concentrated on high-vasopressors treatments. In contrast, PO-CRM and value-based methods, take treatments that are different from the logged clinician decisions. This suggests these policies may be overfitting to avoid the contexts with high SOFA in the distribution. However, some patients arrive with high SOFA scores, so a policy must have a suitable policy to support such individuals, which our method helps to ensure.

While these results are promising, before deploying any offline reinforcement learning algorithm in a high stakes clinical setting, further investigation and collaborations with clinicians would be essential.

## 8 CONCLUSION & FUTURE WORK

To conclude, we identify a novel overfitting challenge that arises when using importance sampling as part of an offline policy learning objective. In particular, the objective can result in a policy that under-weigh certain (lower performing) initial contexts, to achieve higher average value. To address this, we constrain the policy class to only consider logged actions taken by nearby samples. This can be viewed as a similar pessimism constraint that has been used in MDP offline policy learning, but now developed for a non-Markov, direct policy search setting. Our approach yielded strong performance relative to state-of-the-art prior approaches in a tumor growth simulator and a real world dataset on ICU sepsis treatment. Our method may be particularly useful for many applied settings such as healthcare, education and customer interactions, which have a short or medium length decision horizon, but are unlikely to be Markov in the observed per-step variables.

A reader might wonder if a similar benefit might be possible if the algorithm first learned an empirical behavior policy from the logged dataset, and then constraining the policy class to be close to the empirical behavior policy. Our current results suggest that such an approach is likely to require additional innovation: indeed the PO-$\mu$ baseline for the sepsis dataset uses an empirical behavior policy, and does not perform as well as our approach. This is interesting because prior work has shown that using such an empirical behavior policy can yield better off-policy estimates than using true behavior policy (Xie et al., 2018; Hanna et al., 2021) settings, and lead to benefits in off policy learning for contextual bandits Xie et al. (2018). However, this prior work did not consider constraining the policy class nor performing offline RL, and there are some significant subtleties in learning an empirical behavior policy for large continuous spaces. For example, Hanna et al. (2021) demonstrated that learning a behavior policy that maximized the likelihood of the logged data in a IS-based estimator did not yield the most accurate estimate. An interesting direction for future work is whether different ways of learning an empirical behavior policy might yield similar or additional benefits to our locally constrained approach proposed.

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

## A    COUNTER EXAMPLES IN RL SETTINGS

In the main text, we gave an example about the overfitting issue in contextual bandits with large state and action space in small datasets. Here we show that it is even easier for this to occur in sequential reinforcement learning settings, even when only 2 actions are available in the next two examples with or without state aliasing.

**Example 2.** *Consider a sequential treatment problem as shown in Figure 2a. There are two actions available in each state. From the first state, action $a_1$ has a 50% chance of leading to an immediate terminal positive reward $r = 1$ and a 50% chance of leading to an immediate terminal negative reward $r = -1$. From the first state, action $a_2$ also has 50% chance of leading to an immediate terminal positive reward $r = 1$. For the other 50% of states, action $a_2$ results in transitions to additional states, which are followed by additional actions, for another $H - 1$ steps; however, all transitions eventually end in a large negative outcome (e.g. $r = -5$). For example, one could consider a risky surgical procedure that results in many subsequent additional operations and but is ultimately typically unsuccessful. Assume the behavior policy is uniform over each action, yielding $\mu(a = 0|x_1) = \mu(a = 1|x_1) = 0.5$ and a probability of each action sequence following $a_2$ of $\frac{1}{|A|^{H-1}}$. With even minimal data the value of $\pi(x_0) = a_1$ will be accurately estimated as 0. However, when $H$ is large relative to a function of the dataset size, there always exists a action sequence after an initial selection of $a_2$ that is not observed in the dataset. This means that a policy $\pi_2$ that starts with $\pi(x_0) = a_2$ and then selects an unobserved action sequence will essentially put 0 weight on the resulting contexts that incur $r = -5$ outcomes, even though such outcomes will occur 50% of the time after taking action $a_2$. In this case, the value of $\pi_2$ will be overestimated significantly by IS or self-normalized IS. Thus the offline policy optimization will prefer taking action 2 at the first step as a result of overfitting even though the true value of first taking $a_2$ is $-1.5$ and the optimal policy value is 0, obtained by taking action $a_1$.*

Now we add a slight change in the transitions shown in Figure 2a. We can see that model/value-based approach will also fail.

**Example 3.** *In this example, we add another action in the first step. The action 3 and action 1 will lead to the same next state. However in the next state, no matter which action taken, the reward will depends on the action taken in the last step: If $a_1 = 1$, then we have the same reward for $a = 1$ in the example in Figure 2a. If $a_1 = 3$ then we have a reward $-5$. Thus model and value based method will mix the reward for $a_1 = 1$ and $a_1 = 3$ so fail in this example. Other method is not affected by the additional structure as it only add an action with minimum reward.*

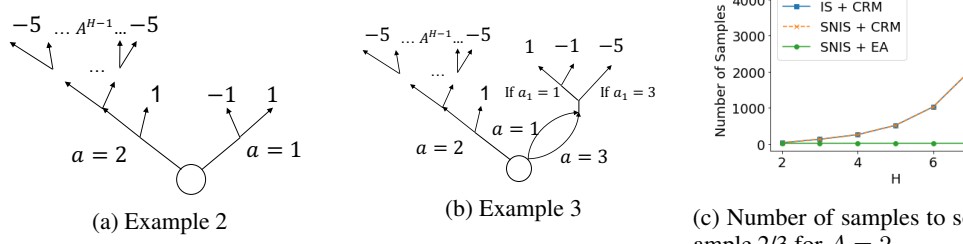

(a) Example 2          (b) Example 3          (c) Number of samples to solve Example 2/3 for $A = 2$.

## B    PROOFS FOR SECTION 5

Proof of Theorem 1

*Proof.*

$$\sum_{x_h^{(j)} \in \mathcal{B}(x_h^{(i)}, \delta)} \frac{\pi(a_h^{(j)} | x_h^{(j)})}{\mu(a_h^{(j)} | x_h^{(j)})} \geq \sum_{x_h^{(j)} \in \mathcal{B}(x_h^{(i)}, \delta)} \pi(a_h^{(j)} | x_h^{(j)}) \tag{8}$$

$$\geq \sum_{x_h^{(j)} \in \mathcal{B}(x_h^{(i)}, \delta)} \max\{0, \pi(a_h^{(j)} | x_h^{(i)}) - \delta L\} \tag{9}$$

$$\geq \sum_{a \in A_h(x_h^{(i)}; \mathcal{D}, \delta)} \max\{0, \pi(a | x_h^{(i)}) - \delta L\} = 1 - \delta L |\mathcal{A}| \tag{10}$$

$\square$

Proof of Corollary 1

*Proof.*

$$\sum_{x_1^{(j)} \in \mathcal{B}(x_1^{(i)}, \delta)} \frac{\max\{W^{(i)}, M\}}{\sum_{i=1}^{n} \max\{W^{(i)}, M\}} \geq \sum_{x_1^{(j)} \in \mathcal{B}(x_1^{(i)}, \delta)} \frac{\max\{W^{(i)}, M\}}{nM} \tag{11}$$

$$\geq \frac{\max\{\sum_{x_1^{(j)} \in \mathcal{B}(x_1^{(i)}, \delta)} W^{(i)}, M\}}{nM} \geq \frac{1 - \delta L |\mathcal{A}|}{nM} \tag{12}$$

$\square$

Proof of Proposition 1

*Proof.* This is due to $\pi(a | x_h^{(i)})$ and $\mu(a | x_h^{(i)})$ are independent from history given $x_h^{(i)}$. So $W_{1:h}^{(i)}$ and $W_h^{(i)}$ are conditionally independent given $x_h^{(i)}$. $\square$

Proof of Theorem 2

*Proof.* We first prove the first part.

Let $P_h(x; \mu)$ to be the distribution of context at $h$-th step with roll-in policy $\mu$. For any fixed $a$, we can define the distribution $P_h(x | a; \mu) = \mu(a | x) P_h(x; \mu) / \sum_a \mu(a | x) P_h(x; \mu)$. For $a$ such that $\mu(a | x) > 0$, $P_h(x | a; \mu)$ is also greater than zero. All $x_h^{(i)}$ with $a_h^{(i)} = a$ are i.i.d. samples draw from the distribution $P_h(x; \mu)$. By the property of nearest neighbor (Cover & Hart, 1967), with probability 1

$$\min_{x_h^{(i)} s.t. a_h^{(i)} = a} \text{dist}(x, x_h^{(i)}) \to 0 < \delta$$

That means with probability 1 $a \in A_h(x; \mathcal{D}, \delta)$ for all $a$ such that $\mu(a | x) > 0$. Given the overlap assumption, we have $a \in A_h(x; \mathcal{D}, \delta)$ for for all $a$ such that $\pi(a | x) > 0$ with probability 1. Thus the solution to Equation 6 is the same as $\arg \max_\pi J(\pi, \mathcal{D}) = \hat{\pi}_{J, \mathcal{D}}$.

Now we prove the second part. By the condition that $M \to \infty$ and $\frac{M}{n} \to 0$ as $n \to \infty$, we have that the truncated IS estimator is mean square consistent (Ionides, 2008):

$$\frac{1}{n} \sum_{i=1}^{n} \left( \sum_{h=1}^{H} r_h^{(i)} \right) \min \left\{ \prod_{t=1}^{H} W_h^{(i)}, M \right\} \xrightarrow{q.m.} v^\pi, \tag{13}$$

as $n \to \infty$. Similarly, we have that the mean of weights converge to 1 in quadratic mean:

$$\frac{1}{n} \sum_{i=1}^{n} \min \left\{ \prod_{t=1}^{H} W_h^{(i)}, M \right\} \xrightarrow{q.m.} 1 \tag{14}$$

Figure 3: The process of hyper-parameters search and test in the experiment.

By continuous mapping theorem, we have that the self-normalized truncated IS converge to $v^\pi$ in probability $\hat{v}_{\text{SNTIS}} \xrightarrow{p} n$. The empirical variance penalty, also converge to 0 almostly surely, since $M/n$ converge to 0:

$$\frac{\sum_{i=1}^{n} \left(r^{(i)} - \hat{v}_{\text{SNTIS}}\right)^2 (\min\{W^{(i)}, M\})^2}{\left(\sum_{i=1}^{n} \min\{W^{(i)}, M\}\right)^2} \leq \frac{M^2}{\left(\sum_{i=1}^{n} \min\{W^{(i)}, M\}\right)^2} \xrightarrow{q.m.} 0 \tag{15}$$

Thus the objective function $J(\pi; \mathcal{D})$ converge to $v^\pi$ in probability:

$$\Pr\left(|J(\pi; \mathcal{D}) - v^\pi| > \epsilon\right) = \delta_n \to 0 \tag{16}$$

Since we assume $|\Pi| < \infty$, we have

$$\Pr\left(\forall \pi \in \Pi \,|\, J(\pi; \mathcal{D}) - v^\pi| > \epsilon\right) = |\Pi|\delta_n \tag{17}$$

So with probability $|\Pi|\delta_n$, for any $\epsilon$:

$$v^{\hat{\pi}_{J,\mathcal{D}}} \geq J(\hat{\pi}_{J,\mathcal{D}}, \mathcal{D}) - \epsilon \tag{18}$$
$$\geq J(\pi^\star, \mathcal{D}) - \epsilon \tag{19}$$
$$\geq v^{\pi^\star} - 2\epsilon, \tag{20}$$

where $\pi^\star$ is $\arg\max_{\pi \in \Pi} v^\pi$. As $|\Pi|\delta_n \to 0$, we proved the true value of empirical maximizer $v^{\hat{\pi}_{J,\mathcal{D}}}$ converge to the maximum of value $\max_{\pi \in \Pi} v^\pi$ in probability. $\square$

## C  EXPERIMENT DETAILS

For all experiment, we follows the 3-phases pipelines to decide the test score we finally report in the paper. The pipeline is described in the main text and summarized in Figure **??** here as well.

### C.1  EXPERIMENT DETAILS IN TGI SIMULATOR

The TGI simulator describes low-grade gliomas (LGG) growth kinetics in response to chemotherapy in a horizon of 30 months using an ordinary differential equation model. The parameter in ODEs are estimated using data from adult diffuse LGG during and after chemotherapy was used, in a horizon of 30 months. The goal in this environment is to achieve a reduction in mean tumor diameters (MTD) while reducing the drug dosage (Yauney & Shah, 2018). We includes the MTD, the drug concentration, and the number of month (time-step) in the context space. Notice that this context space is non-Markov as it does not include all parameters in the ODEs. Actions are binary representing taking the full dose or no dose which is same as prior work (Yauney & Shah, 2018). The reward at each step consist of an immediate penalty proportional to the drug concentration, and a delayed reward at the end measures the decrease of MTD compared with the beginning. Each episodes, the parameters including the initial MTD are sampled from a log-Normal distribution as (Ribba et al., 2012) representing the difference in individuals. The behavior policy is a fixed dosing schedule of 9 months (the median duration from Peyre et al. (2010)) plus $30\%$ of a uniformly random choice of actions. We run all algorithms on a training set with 1000 episodes with different hyperparameters (listed below), and 5 restarts, saving checkpoints along the training. Then we select the best policy for each algorithm by $\hat{v}_{\text{SNTIS}}$ on the validation set with 1000 episodes as well.

**Hyperparameters** In the first part of Table 3 we show the searched hyperparameters of each algorithm, except that the parameter $b$ in PQL is set adaptively as the 2-percentile of the score on the training set as in the original paper Liu et al. (2020). As we know the behavior policy, we use the true behavior policy in BCQ and PQL algorithm. So BCQ threshold takes only two values as the behavior policy is $\epsilon$-deterministic so there are only two distinct values. In the second part of Table 3 we specify some fixed hyperparameters/settings for all algorithm. All policy/Q functions are approximated by fully-connected neural networks with two hidden layers with 32 units.

| Hyperparameters | used in algorithms | values |
|---|---|---|
| $\delta$ | POELA | $0.05, 0.1, 0.5$ |
| CRM Var coefficient | POELA, PO-CRM | $0, 0.1, 1$ |
| BCQ threshold | BCQ, PQL | $0.0, 0.2$ |
| $M$ in $\hat{v}_{\text{SNTIS}}$ | All | 1000 |
| Max training steps | POELA, PO-CRM, PO-$\mu$ | 500 |
|  | BCQ, PQL | 5000 |
| Number of checkpoints | All | 50 |
| Batch size | BCQ, PQL | 100 |

Table 3: Hyperparameters in the TGI simulator experiment

The difference in the max update steps and checkpoints frequency is caused by the fact that BCQ and PQL is updated by stochastic gradient descent and all policy optimization based on SNTIS is using gradient descent.

## C.2 EXPERIMENT DETAILS IN MIMIC III DATASET

The MIMIC III sepsis dataset is available upon application and training: https://mimic.mit.edu/iii/gettingstarted/. The code to extract the cohort is available on: https://gitlab.doc.ic.ac.uk/AIClinician/AIClinician. This cohort consists of data for 14971 patients. The contexts for each patient consist of 44 features, summarized in 4-hour intervals, for at most 20 steps. The actions we consider are the prescription of IV fluids and vasopressors. Each of the two treatments is binned into 5 discrete actions according to the dosage amounts, resulting in 25 possible actions. The rewards are defined from the 90-day mortality in the logs, 100 if the patient survives and 0 otherwise.

We now provide details of the experiment on MIMIC III sepsis dataset here. We run all algorithms on a training set with 8982 trajectories with different hyperparameters (listed below), and 3 restarts, saving checkpoints along the training. Then we select the best policy for each algorithm by $\hat{v}_{\text{SNTIS}}$ on the validation set with 2994 trajectories. Finally we get the $\hat{v}_{\text{SNTIS}}$ evaluation on the test set with 2995 trajectories.

In the first part of Table 4 we list the hyperparameters that we searched on the validation set for each algorithm, except that the parameter $b$ in PQL is set adaptively as the 2-percentile of the score on the training set as in the original paper Liu et al. (2020). In the second part of Table 3 we specify some fixed hyperparameters/settings for all algorithm. All policy/Q functions are approximated by fully-connected neural networks with two hidden layers with 256 units.

As we explained, the difference in the max update steps and checkpoints frequency is caused by the fact that BCQ and PQL is updated by stochastic gradient descent and all policy optimization based on SNTIS is using gradient descent.

### C.2.1 ACTION VISUALIZATION FOR MID/LOW-SOFA PATIENTS

In the main text we show the action visualization for high-SOFA ($> 15$) as a diagnose of how algorithm behave for the high-risk patients. For the completeness of results here we show the visualization of mid-SOFA ($5 - 15$) and low-SOFA ($< 5$) patient contexts.

| Hyperparameters | used in algorithms | values |
|---|---|---|
| $\delta$ | POELA | $0.4, 0.6, 0.8, 1.0$ |
| $\hat{\mu}$ threshold | PO-$\mu$ | $0.01, 0.02, 0.05, 0.1$ |
| CRM Var coefficient | POELA, PO-CRM | $0, 0.1, 1, 10$ |
| BCQ threshold | BCQ, PQL | $0.0, 0.01, 0.05, 0.1, 0.3, 0.5$ |
| $M$ in $\hat{v}_{\text{SNTIS}}$ | All | $1000$ |
| Max training steps | POELA, PO-CRM, PO-$\mu$ | $1000$ |
| | BCQ, PQL | $10000$ |
| Number of checkpoints | All | $100$ |
| Batch size | BCQ, PQL | $100$ |

Table 4: Hyperparameters in the MIMIC III sepsis experiment

### C.2.2 ADDITIONAL ABLATION STUDY: WITHOUT EFFECTIVE SAMPLE SIZE CONSTRAINTS FOR HYPER-PARAMETER SELECTION ON VALIDATION SET

In the main text, we set an effective sample size threshold of 200 for a policy/hyper-parameter to be selected on validation set. This is to make sure we have large enough effective sample size on the test set to provide reliable off-policy test estimates. Here we show the result if we do not threshold the effective sample size on validation set. Generally, all algorithms will prefer a high off-policy estimates without enough effective sample size. On the test set, all algorithms yields a small effective sample size, thus unreliable off-policy estimates and large bootstrap confidence interval. The proposed methods is better than baselines but also has much smaller $95\%$ bootstrap lower bound than with the effective sample size constraint.

| Method | POELA | PO-$\mu$ | PO-CRM | BCQ | PQL |
|---|---|---|---|---|---|
| Test SNTIS | 87.63(86.29) | 82.36 | 82.36 | 83.28 | 96.32 |
| 95% BCa LB | 85.06(83.51) | 64.92 | 63.48 | 56.65 | 57.25 |
| 95% BCa UB | 90.00(88.59) | 94.22 | 93.62 | 100 | 100 |
| Test ESS | 528.18(491.71) | 21.23 | 21.23 | 9.04 | 1.27 |

Table 5: Test evaluation without effective sample size constraint on the validation set, $(0.05, 0.95)$ BCa bootstrap interval, and effective sample size in the sepsis cohort of MIMIC III dataset. Value inside parenthesis of POELA is without CRM variance penalty.

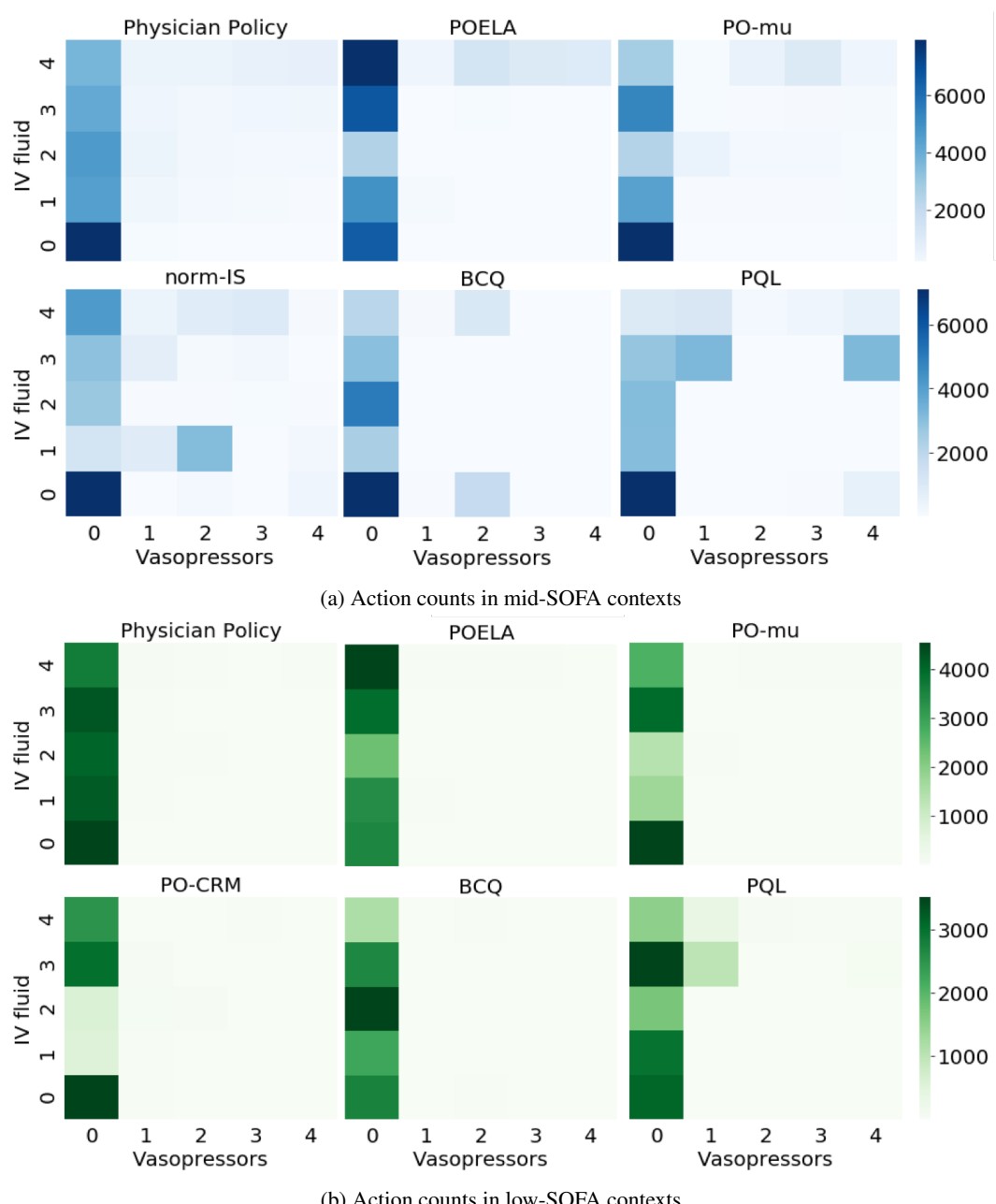

(a) Action counts in mid-SOFA contexts

(b) Action counts in low-SOFA contexts

Figure 4: (a): Action counts heatmap in mid-SOFA contexts of the policy learned from different algorithms. (b): Action counts heatmap in low-SOFA contexts of the policy learned from different algorithms.

## C.3 IMPORTANCE WEIGHTS IN LOW-REWARD TRAJECTORIES

To examine if the proposed overfitting phenomenon exists in real experimental datasets, we compute the importance weights of the learned policy on the low-reward trajectories in the training data for our MIMIC III dataset and our tumor simulator. Our hypothesis is that overfitting of the importance weights in policy gradient methods may result in the algorithm avoiding initial states with low rewards, which motivated our proposed algorithm.

In MIMIC III dataset the reward for a trajectory is either $0$ or $100$. We define the low-reward trajectories as those with $0$ reward. Low-reward trajectories are over $60\%$ of all trajectories in the

dataset. In the Tumor simulation experiment we define a low-reward trajectory when reward is less than $-2$. Over $95\%$ of trajectories in the Tumor simulation dataset are low-reward trajectories.

The table below shows, for each algorithm and setting, the sum of the SNTIS weights of the learned policy on the training set, for low-reward trajectory states. Our primary interest is to illustrate that alternate policy gradient methods that are also suitable for non-Markov domains, can exhibit the importance sampling overfitting of avoiding low reward trajectories. We indeed see that our POELA method has a much larger weight on low-reward trajectories than these alternate offline policy search methods:

| Method | POELA | PO-$\mu$ | PO-CRM |
|---|---|---|---|
| MIMIC III | 0.028 | 0.001 | 0.003 |
| Tumor non-MDP | 0.054 | - (fixed policy) | 0.005 |

Table 6: Importance weights overfitting: sum of SNTIS weights of learned policy on the training set.

The Q-learning baselines we consider (BCQ and PQL) do not directly use the importance weights, but they do try to avoid actions and/or states and actions with little support. Our POELA method can be viewed as being similarly inspired, but for non-Markovian settings where policy gradient is beneficial. We also compute the SNTIS weights of the BCQ/PQL policy on the training set in the Markov domain that satisfies the Markov assumptions of BCQ/PQL. Here we can see that our POELA, BCQ and PQL all still give significantly more weight to low reward trajectories than the alternate policy gradient methods:

| Method | POELA | PO-$\mu$ | PO-CRM | BCQ | PQL |
|---|---|---|---|---|---|
| Tumor MDP | 0.097 | - (fixed policy) | 0.0004 | 0.083 | 0.124 |

Table 7: Importance weights overfitting: sum of SNTIS weights of learned policy on the training set.

These results help illustrate that the over avoidance of low-reward trajectories can be observed by past policy gradient methods in our datasets. Of course, one challenge is that in real settings, an excellent policy may have low importance weights in avoidable low-reward states and trajectories, but should have higher importance weights in non-avoidable low reward starting states and trajectories. To get a fuller picture of performance, it is helpful to look both at the weights on trajectories with low rewards and the test evaluation results. Compared with strong policy gradient baselines, our proposed regularization method have larger importance weights on low-reward trajectories, and the gap between training/validation evaluation and online test performance is also smaller, suggesting that we are less likely to learn policies that erroneously believe they can avoid unavoidable low reward settings.

## C.4 THE EFFECT OF ELIGIBLE ACTION CONSTRAINTS $\delta$

In this section we explore how the choice of $\delta$, which constrains the policy class through impacting the eligible actions, impacts empirical performance. Larger $\delta$ corresponds to a less constrained policy class. Other hyper-parameters are selected by the same procedure as described in previous sections.

As $\delta$ increases, the policy search operates with less constraints. The results show that in this case, our policy gradient method produces a policy with a higher value in the training set, but that policy may not perform as well in the test evaluation, and may have a smaller effective sample size than when a smaller $\delta$ is used. The best hyper-parameter value $\delta$ lies in the middle of the explored range. $\delta$ can be selected based on performance and effective sample size.

| $\delta$ | 0.4 | 0.6 | 0.8 | 1.0 |
|---|---|---|---|---|
| training $\hat{v}_{\text{SNTIS}}$ | 91.62 | 98.41 | 98.9 | 99.12 |
| training ESS | 3601.12 | 2242.07 | 1993.08 | 1769.46 |
| test $\hat{v}_{\text{SNTIS}}$ | 86.62 | 90.07 | 91.46 | 90.23 |
| test ESS | 1278.08 | 819.64 | 624.92 | 542.53 |

Table 8: The effect of eligible action constraints $\delta$ on the results in MIMIC III sepsis dataset.

