# OpenReview forum: "Avoiding Overfitting to the Importance Weights in Offline Policy Optimization"
_ICLR.cc/2022/Conference — ICLR 2022 Submitted_

### Official Review · Reviewer_2pVL · 2021-10-19

**Correctness:** 4
**Technical Novelty And Significance:** 3
**Empirical Novelty And Significance:** 2
**Recommendation:** 6
**Confidence:** 3

**Main Review:**

**Strengths**
- the paper tackles the important and relevant issue of overfitting in counterfactual risk minimization. the motivation is also nicely supported with some toy examples.
- the paper identifies the source of overfitting that is not addressed in previous works.
- the paper proposed a novel algorithm to address the overfitting issue with some gurantees
related work is covered rigorously
- the code is provided as supplementary material. I encourage the authors to publicize it upon publication to ensure the reproducibility

**Weaknesses**
- I skimmed through the proofs of Theorem 1 and Corollary 1. They look correct, but the bounds seem very loose and might be meaningless. It would be more interesting to see how the empirical sum of weights are different among different methods using synthetic data.
- All experiments are based on real-world data, meaning that the policy performances are now measured with OPE on a test set, which might not be accurate. Having synthetic experiments as a complement would strengthen the empirical contribution.
- Do we still observe the overfitting issue when we use k-fold cross validation?


**Summary Of The Paper:**

The paper studies the overfitting issue in counterfactual policy learning. The paper first  identifies an important overfitting phenomenon in optimizing the importance weighted return, and proposes an algorithm to avoid the issue. The notable limitation of some previous approaches is that they use penalization at the policy level and do not directly address the problem with avoiding contexts with low reward. The proposed POELA method addresses deficient support in the observed finite data rather than in the expected behaviors.  Some experiments show the benefit of the proposed approach.

**Summary Of The Review:**

The paper identifies the unexplored aspect of the overfitting issue in off-policy learning. The motivation was easy to follow with some easy examples. I would recommend a weak accept at this moment. Some additional efforts in the experiments (as I described above) would strengthen the contribution more.

---

> ### Author Response · Authors · 2021-11-23
> **Response**
>
> We thank the reviewer’s constructive comments. See the following response to each of the major comment points. Updates to the paper including new experimental details are highlighted in red in the pdf.
> 1) According to the reviewer’s suggestion, we updated the paper to include the importance weights in the low-reward trajectories in the simulation and MIMIC experiments in Appendix C.3. We do want to emphasize that the splitting of low-reward/high-reward uses the reward information. Unlike the illustrative example, in this case, we cannot know that if it really learns a better decision or just avoids a particular state which should not be affected by no matter what decisions. In fact, a really good policy will also have very low importance weights in the low-reward samples. These results need to be combined with the test evaluation results to understand the overfitting phenomenon.
> 2) The numerical simulation in section 7.1 and Appendix A is done on a simulated domain. They include online testing results in the simulator.
> 3) Yes, cross-validation can often be used to reduce overfitting issues in machine learning. However, there are two unique challenges in batch RL. First, the cross-validation itself is using (high variance) off-policy evaluation metrics and is more vulnerable than in supervised learning. So learning a policy more robust to potential overfitting is still valuable. Second, the overfitting can be caused by two reasons, one is limited data as the same as in standard supervised learning and can be addressed by cross-validation. The overfitting risk pointed in this paper is caused by the nature of the importance sampling estimator. The issue still bothers cross-validation if it is using importance sampling based approach and the number of model hyper-parameter/checkpoints is large.

---

### Official Review · Reviewer_qYyg · 2021-10-28

**Correctness:** 3
**Technical Novelty And Significance:** 3
**Empirical Novelty And Significance:** 3
**Recommendation:** 6
**Confidence:** 3

**Main Review:**

Major comments:

* In Section 3, the author(s) decomposed the importance sampling estimator into three parts and argued that the second term can be problematic. I suggest the author(s) to design a toy simulation example to evaluate the order of magnitude of these three error terms (the first error term being $E_{\hat{p}} v^{\pi}(x) - E_{p} v^{\pi}(x)$ where $p$ corresponds to the oracle context distribution).

* In Equations (3) and (4), suppose we plug-in the empirical behavior distribution for $\mu$, then $\hat{p}(x)$ equals $W(x)$. This would solve the over-fitting problem. In practice, according to the semiparametric theory, even if the true importance sampling ratios are known to us, the resulting estimator with an estimated ratio would be more efficient (see e.g., Tsiatis, 2006, Semiparametric Theory and Missing Data). Would you please clarify why you did not use such a simple method with an estimated importance sampling ratio?

* Would you please elaborate the constraint in Equation (6)? Why would such a constraint help solve the offline learning problem?

* The condition $|\Pi|<+\infty$ is not sufficient to guarantee the validity of Theorem 2. As shown in the proof, $|\Pi| \delta_n$ shall decay to zero. The author(s) needs to provide an upper bound for $\delta_n$ in the proof (as a function of M and $n$) and presents the corresponding condition for $|\Pi|$. In addition, naively apply the Bonferroni's inequality would give a loose bound. Concentration inequalities would be preferred to relax the resulting condition for $|\Pi|$.

* The proposed algorithm relies on several tuning parameters, such as $\delta$, $M$, $\lambda$ and $\alpha$. Is your algorithm sensitive to the choice of these tuning parameters? How would you recommend practioners to select these hyper-parameters? Could cross-validation be possibly employed? In addition, what policy class do you consider in the numerical experiments?

* Just curious how the proposed compares with existing baselines in standard openAI Gym environment. Have you conducted some related experiments?

* Since the numerical experiments use policy evaluation algorithms to compare different policies, I wonder if the estimated values are sensitive to the choice of the policy evaluation algorithm or the tuning parameters used in the algorithm. Would you please discuss?

Minor comments:

* Page 3, Line -12. Shall $hatp(x)$ be $\hat{p}(x)$?
* Page 4, Lines -17 and -18. Missing parentheses in $E [ ... ]$.
* Page 6, Theorem 2. $M$ as a function of $n$?
* Page 13, Line -14. "to be" -> "be".

**Summary Of The Paper:**

The paper considers offline policy optimization. The author(s) discussed the issue of overfitting of the importance weights in existing offline algorithms and developed an algorithm to alleviate the issue. Results are supported via theories and real datasets from healthcare applications.

**Summary Of The Review:**

My detailed comments are given under the "Main Review" section. I have some concerns about the motivation of the method (#2), the theoretical results (#4), the choice of the tuning parameters (#5) and the sensitivity of the estimated values in the empirical studies to the choice of the policy evaluation algorithms (#6). So I give a score of 5. However, I would like to increase my score shall my comments be addressed.

---

> ### Author Response · Authors · 2021-11-23
> **Response**
>
> We thank the reviewer’s constructive comments. See the following response to each of the major comment points.
> 1) We thank the reviewer’s suggestion. The toy example in the main paper (Example 1) and Appendix (Example 2 and 3) can both show this point. In Example 1, the third term is zero, the first term is a standard concentration error, for which even full information (knowing the whole reward function) supervised learning algorithm will also suffer. The only error that IS based suffers particularly in Example 1 is the second error term.
> 2) Thank you for asking about this alternative. There are several reasons we did not pursue this direction. A key issue is that in continuous state spaces,
> An estimated behavior policy or ratio itself will have more practical difficulty. First, in continuous state space, p_hat(a|s) will almost always be 1 since we will only observe one state once mostly. So simply using the empirical frequency and making the error term disappear is not always a good choice for the whole learning process. Second, generally estimating p_hat(a|s) as a conditional density/mass function is hard in continuous case. This will further introduce another error term which is the difference between w_hat (estimated IS ratio) and w. We doubt similar error decomposition and overfitting can be observed there. Third, the estimated behavior policy or IS ratio itself has different amounts of uncertainty depending on the data distribution. A single estimated value can not fully characterize this uncertainty which actually causes the overfitting. The proposed method is to address this. We will add a more detailed version of this discussion in a later version of the paper. In fact, in the MIMIC experiment, we compared with this method, referred to as “PO-\mu”, since the behavior policy is always estimated in all methods.
> 3) Constraint in Equation 6. Thank you for allowing us to clarify why this constraint is useful. This constraint ensures that for the observed states in the dataset, the considered target policy must put 100% of its weight on actions in the eligible action set, which are designed to be those with empirical support in the dataset.
> 4) Condition |Pi| <= +infinity sufficiency. Thanks for allowing us to clarify this. Actually, in our proof, we use a fixed $\delta$ instead of a function of $n$. We can see this from Corollary 1 where we only need $\delta$ to be small enough such that the $1 - \delta L |A| > 0$. $n$ does not need to decrease with n. Thanks for your point about the Bonferroni inequality: we completely agree that the resulting condition for |Π| can be relaxed by using tighter analysis.  This was not the focus of our current work but it is definitely possible to easily replace this with other function class complexity measurements.
> 5) Hyperparameters: Thanks for asking: in Appendix C we discuss the range of hyperparameters considered: they are selected based on the performance on a validation set, which we would also recommend to practitioners.  We are using fully connected neural networks in our experiments: in MIMIC we use two hidden layers with 256 units and in the tumor simulation we use two hidden layers with 32 units.
> 6) Comparison to Gym baselines. We are motivated by real-world datasets like MIMIC III where it’s hard to verify the Markov assumption. We think some of the MuJoCo or other Gym baselines are not the most suitable for the problem scenario we want to solve. We agree that the policy gradient method may not have comparable performance compared to Q learning or actor-critic when the horizon is longer than hundreds and the Markov assumption is appropriate.
> 7) Impact of tuning parameters. In Figure 1a we consider the result of different methods with different hyperparameter settings. There is quite a bit of variation across all the methods as a function of the hyperparameters used though our POELA is more consistently at the best frontier of balancing performance and effective sample size.
>
> Thank you for the other additional comments which we will address in a revised version.

---

### Official Review · Reviewer_oG3M · 2021-10-31

**Correctness:** 4
**Technical Novelty And Significance:** 2
**Empirical Novelty And Significance:** 3
**Recommendation:** 5
**Confidence:** 4

**Main Review:**

I find the proposed method of this paper is quite intuitive and the baselines compared seem to be appropriate. The Example 1 is good to understand with more motivating examples in the appendix. I also love the clinically-inspired evaluations.

My concerns are as followed:
1. Regarding the methods
  - A) This phenomenon seems to only apply when highly negative reward are present. One way is to shift and scale the rewards to be positive, and capped the importance weights to avoid over-maximize the reward. I am curious if authors can compare with this method, or at least illustrate why this way of shifting reward will not work.
  - B) The distance considered in the paper uses the euclidean distance in the raw input space, which may not work when lots of missingness present in the data, or in settings when some features of the inputs are not relevant to the reward. I think learning a meaningful latent space that depends on the reward like [1] could further improve this method.
  - C) The novelty of this work is not high as similar ideas have been proposed but not particularly in the non-MDP setting as acknowledged by the authors.


2. Regarding the experimental results
  - A) Although it's intuitive that this overfitting phenomenon could happen theoretically, only qualitative figures in Fig. 1(b) support this claim. I believe it's crucial to support this statement in the experiments. I think authors can quantify it in both simulations and the MIMIC3 by showing that the importance weights are close to 0 for the baselines in lower-reward states.
  - B) Since you use WIS as the validation metric, it might favor methods also optimized under IS-based methods but might give an disadvantage to non-IS based methods like PQL. Though I understand you have to choose a metric.
  - C) In Fig. 1(a) why there is no blue point around x=200~400? Just want to see if blue actually is better in those regions.

3. Regarding the presentations (this has little effect on my evaluation)
  - A) Bolding table 1 and 2 can help readers quickly understand which method is better
  - B) In Sec. 3 please list the source of equation (2).

[1] Zhang, Amy, et al. "Learning Invariant Representations for Reinforcement Learning without Reconstruction." International Conference on Learning Representations. 2020.

**Summary Of The Paper:**

When optimizing an policy under batch data under importance weighting adjustment of batch data, this paper observes that methods can learn to ignore states with highly negative rewards by having lower support to the behavior policy. And this paper proposes to solve this problem by only constraining actions that have observed in the nearby states in the batch data. Although various similarly ideas have been proposed like regularizing toward behavior policy, this paper improves on importance sampling methods that does not assume MDP that separates itself from recent baselines like PQL. They also show some analysis on this problem. They show that compared to recent baselines they improve in a tumor growth simulation environment when MDP assumption is violated, and a real-world sepsis dataset.

**Summary Of The Review:**

Pros:
+ The method is easy and simple and shows improvement in both experiments that include non-MDP simulations and a real-world dataset
+ The writing is mostly clear
+ The improvement shown in a real-world clinical environment is encouraging.

Cons:
- No quantitative analysis if the overfitting actually happens.
- Simple heuristics like shifting rewards to be positive and avoiding over-maximizing reward is not compared
- The novelty of this work seems limited since similar ideas have been proposed.
- The distance considered in this paper may not handle missingness or when some input features are irrelevant to the reward.

Although I feel the proposed overfitting phenomenon can be an interesting contribution, the authors should quantify if it happens in the experiments and justify their method by comparing with heuristics like shifting the reward. I think addressing these can further push the paper over the acceptance bar.

---

> ### Author Response · Authors · 2021-11-23
> **Response**
>
>
> We thank the reviewer’s constructive comments. See the following response to each of the major comment points. Updates to the paper including new experimental details are highlighted in red in the pdf.
>
> (A) Is shifting all rewards to be positive sufficient: Thank you for this question: indeed this is something we considered during our earlier work and it can be sufficient if one wishes to restrict consideration only to using importance sampling estimators with clipping. Unfortunately it is not sufficient when we wish to use better policy evaluation estimators that typically leverage different forms of a control variate, such as a baseline function, doubly robust estimator, or weighted importance sampling (as a multiplicative control variate). Under these methods, the reward will be re-centered around zero and any constant shift does not affect the algorithm’s solution. This is also verified in our MIMIC III experiments where the rewards are all positive. As a more minor additional point, in settings where the trajectory length is variable, a constant shift in reward may change the dense reward to a more sparse and delayed reward signal, for example, a CartPole or MountainCar domain. For all these reasons we wanted to go beyond shifting to positive rewards.
>
> (B) More sophisticated latent space: Thank you for the interesting suggestion.  We did not focus on this direction but strongly agree with the reviewer on learning a meaningful metric space could further improve this method. We will add this to the discussion.
>
> (C) Relation to work in the MDP space. While we agree that there is related work on handling errors due to bootstrapping and lack of support in the MDP setting, this prior batch RL and bandit work has not described this overfitting phenomenon and its cause -- the ability to manipulate the “effective” initial context distribution.
>
> Experimental results A, Overfitting in experiments: Thank you for this good suggestion. Given your suggestion we have updated the paper to include the importance weights in low-reward trajectories in the simulation and MIMIC experiments in Appendix C.3 and an additional discussion. We can see that our approach places more weight on these than prior policy gradient approaches. We have included an additional discussion of this and the importance of also considering the performance results.
>
> Experimental results B, WIS as metric: We agree that the choice of WIS may affect the evaluation for different methods. We choose WIS instead of fitted Q evaluation since it is more robust to the Markov and other types of function approximation assumption, which is generally not true or unverifiable in the MIMIC dataset. Note that for the simulations we use the true on-policy test results in the domain, which avoids this need to pick a particular estimator for the policy evaluation.
>
> Experimental results C, blue point: Thanks for the interesting observation. We searched a range of hyperparameters for the proposed and baseline methods: see the appendix for the details of hyperparameters’ range. We notice that the proposed method either has an effective sample size > 200 (when we turn on the constraints) or a very small effective sample size (when the constraints are turned off by the hyperparameter). To us this suggests the benefit of having the constraints in that it will ensure a decently large effective sample size. We will clarify this discussion in the text.
>
> Thank you for the additional presentation suggestions which we will address.

---

### Official Review · Reviewer_EAaa · 2021-11-06

**Correctness:** 3
**Technical Novelty And Significance:** 3
**Empirical Novelty And Significance:** Not applicable
**Recommendation:** 5
**Confidence:** 4

**Main Review:**



- One major issue with the proposed approach is the expressivity of the policy class. Although authors in Theorem 2 have shown that asymptotically, the expressivity stays the same, one question here is that what really happens in the non-asymptotic setting and how much we loose expressivity with the introduced constraint.

- In other words, the reviewer believes one missing part of this paper is to show how much local constraints on actions (eq. 5) is hurting the performance in the non-asymptotic setting. How does this loss change as sample size grows? Showing empirical results even on synthetic datasets will be helpful.

- One issue with Table 1 is that authors say their approach does not make any Markovian assumption and hence, its performance is worse than other competitors in the MDP setting while it outperforms other methods in the non-MDP setting. One question here is that is this comparison fair? In other words, do other methods make Markovian assumption and at the same time we expect them to perform well in a non-MDP setting?

- The writing of the paper needs a major review. Examples are but not limited to:

Page 1, Introduction Section: “different than”
Page 2, line 2: : “learning an policy”
Page 2, Section 2, Line 6: “over next context”
Page 2, Section 2: “also sometimes refered”
Page 3, Section 3: “but a similar results”
Page 3, Section 3: “for unnormalized estimator.”
Page 5, Section 4: “function class is closed respect”
Page 7, Section 6: “Euclidean distance in over”
Page 7, Section 7: “details are provided are in the Appendix”
Page 7, Section 7: “The reward consist of”
Page 9, Section 8: “in a policy that under-weigh certain (lower performing) initial contexts”


**Summary Of The Paper:**

In this paper, authors have introduced an overfitting phenomenon that has not been addressed in previous works related to policy optimization and importance sampling. They then propose a new constraint on policy and a new algorithm which avoids the overfitting. They have provided theoretical justification on why their proposed method works and show experimental results to show the effectiveness of their approach.

**Summary Of The Review:**

This paper introduces a legit overfitting issue and proposed a solution for it. However, the reviewer is not convinced enough about the effectiveness of the proposed approach due to the concerns raised in the main review plus the fact that the paper needs a major writing revision.

---

> ### Author Response · Authors · 2021-11-23
> **Response**
>
> We thank the reviewer for his/her constructive comments which we will address in the revised draft. Here we focus on the key questions raised
>
> 1) Policy class expressivity. We agree that it is interesting to explore how these constraints on the policy class impact performance in the finite sample regime. While our initially provided experimental results show that including this constraint is beneficial in the finite sample regime, we  have added additional results for how different values of $\delta$ (which constrain the eligible actions) affect the resulting policy and test performance for a fixed dataset size in MIMIC III, in Appendix C (see text in red). Empirical performance (on the test set) is consistently strong, and better than prior methods shown in Table 2, across a wide range of deltas (0.6-1) but the exact best performance does vary, as expected, with delta. This is likely to be unavoidable as we are restricting the policy class expressivity to improve generalization performance.  We do highlight that our constraints/regularization is a data-dependent one, and it does not constrain the policy class in parts of the space with dense data.
>
> 2. Table 1 and performance of Markov and non-Markov methods in Markov and non-Markov domains. We completely agree with the reviewer that we do not expect that Markov methods will necessarily do well in non Markovian environments and is one of the reasons that we include both domains. Our aim was to illustrate that methods that assume the domain is a MDP can struggle in settings where the Markov assumption may be violated, and that new methods, like our POELA, that do not make the Markov assumption can be beneficial. In general it is not trivial to test from offline data if a domain satisfies the Markov assumption: recent work [1] provides a hypothesis test but only guarantees accuracy when the number of trajectories or the length of a trajectory goes to infinity, and a few additional assumptions.  Therefore our work can be viewed as a more robust approach suitable for still ensuring strong performance even if the Markov assumption is violated, as our experiments demonstrate.
>
> [1] Shi, C., Wan, R., Song, R., Lu, W., & Leng, L. (2020, November). Does the Markov decision process fit the data: testing for the Markov property in sequential decision making. ICML
>
> 3. Thank you for the feedback on writing errors and issues and we will update the text accordingly.

---

### Decision · Program_Chairs · 2022-01-20

**Decision:**

Reject

**Comment:**

In this paper, the authors proposed an offline policy optimization algorithm, motivated by an analysis of the upper bound error of importance sampling policy value estimator. Specifically, by the decomposition of the error in a particularly way, the authors identified some error which does not converge. Then, the authors introduce the contraints over feasible actions to avoid the overfitting induced by such errors. Finally, the authors tested the proposed algorithm empirically.

The paper is well-motivated and the authors addressed some of the questions in their rebuttals. However, there are still several issues need to be addressed,

- The alternative practical estimator with plug-in behavior distribution would perfectly avoid the over-fitting, which is, however, ignored. This is an important and easy-to-implemented competitor.

- The pessimistic principle in the face of uncertainty (PFU) has been exploited extensively in offline policy optimization problem. How the proposed algorithm is connected to the PFU has not been discussed carefully, especially in terms of non-asymptotic sample complexity, which makes the paper is not well-positioned.

- While the motivation is derived from the unbiased importance sampling estimator, the counterfactual risk minimization in Equation 7 is introduced suddently, without clear justification.

- In my opinion, for a better clarification of the paper, the expressiveness of the policy family should not be discussed in this way. I understand the authors would like to avoid any possible degeneration, and explain the asymptotic lossless in terms of policy flexibility. However, the whole point of the paper is trying to introduce some mechanism to avoid the possible overfitting by regularizing the policy family. In other words, the restriction is on purpose and beneficial. I think the argument of policy family expressiveness should be re-considered and re-discussed.


Minor:

- Markovian vs. non-Markovian baseline comparison is not fair, and more comparison on well-known benchmarks, e.g., OpenAI gym, should be conducted.
- The \sigma upper bound should be explicitly provided and verified in practice.

In sum, the paper is well-motivated, however, need further improvement to be pulished.